# Health Status of Afghan Refugees in Europe: Policy and Practice Implications for an Optimised Healthcare

**DOI:** 10.3390/ijerph19159157

**Published:** 2022-07-27

**Authors:** Michael Matsangos, Laoura Ziaka, Artistomenis K. Exadaktylos, Jolanta Klukowska-Rötzler, Mairi Ziaka

**Affiliations:** 1Department of General Surgery, Insel Gruppe AG, Kreditorenbuchhaltung, Freiburgstrasse 18, 3010 Bern , Switzerland; mike.mjr@hotmail.com; 2Department of Special Needs Education, University of Oslo, 0315 Oslo, Norway; laouraziaka@gmail.com; 3Department of Emergency Medicine, Inselspital, University Hospital, University of Bern, 3010 Bern, Switzerland; aristomenis.exadaktylos@insel.ch (A.K.E.); jolanta.klukowska-roetzler@insel.ch (J.K.-R.); 4Department of Internal Medicine, General Hospital of Thun, 3600 Thun, Switzerland

**Keywords:** Afghan refugees, health status, internal conflicts, migration, asylum seekers policy and practice, biopsychosocial factors, epigenetic changes, displacement, war, communicable diseases, non-communicable diseases, women’s healthcare, sexual violence, mental health

## Abstract

Four decades of civil war, violence, and destabilisation have forced millions of Afghans to flee their homes and to move to other countries worldwide. This increasing phenomenon may challenge physicians unfamiliar with the health status of this population, which may be markedly different from that of the host country. Moreover, several factors during their migration, such as transport in closed containers, accidental injuries, malnutrition, and accommodation in detention centres and refugee camps have a major influence on the health of refugees. By taking into account the variety of the specific diseases among migrant groups, the diversity of the origins of refugees and asylum seekers, and the increasing numbers of Afghan refugees, in this review we focus on the population of Afghans and describe their health status with the aim of optimising our medical approach and management. Our literature review shows that the most prevalent reported infections are tuberculosis and other respiratory tract infections and parasitic diseases, for example leishmaniasis, malaria, and intestinal parasitic infections. Anaemia, hyperlipidaemia, arterial hypertension, diabetes, smoking, overweight, malnutrition, low socioeconomic status, and poor access to healthcare facilities are additional risk factors for non-communicable diseases among Afghan refugees. With regards mental health issues, depression and post-traumatic stress disorder (PTSD) are the most common diagnoses and culture shock and the feeling of being uprooted modulate their persistence. Further research is needed in order to provide us with extensive, high-quality data about the health status of Afghan refugees. The main objective of this review is to identify protective factors which could ensure key health concepts and good clinical practice.

## 1. Introduction

All refugees, regardless of their origin, are of international concern. Given that their relationship with their own state has broken down, the global community has a clear responsibility to protect their fundamental rights. Underlying conflicts may threaten all generations, and many migrants have travelled through treacherous and stressful conditions to reach their destination countries.

Afghanistan is a country which has consistently produced immigrants due to its geographical location, ethnic structure and continual internal disturbances. Afghans have been migrating to neighbouring countries for centuries. The most important mass migration movement in recent years occurred when the country was occupied in 1979 [1].

Four decades of civil war and serious human rights abuses have forced millions of Afghan men, women and children to flee their homes and seek refuge in other parts of Afghanistan or outside the country. Since the armed conflict began in 1979, civilians—women and children in particular—have suffered enormously from its devastating consequences [1]. The conditions of migration and settling in the host countries certainly differ between Afghans and may depend on their socioeconomic status, religion status, and affiliation to ethnic minorities.

There has recently been a spike in the emigration of Afghans, especially those seeking asylum in Europe, and this may have been encouraged by the mass exodus of Syrians. Between 2015 and 2017, nearly one million Afghans sought asylum worldwide [2,3]. Nearly two thirds of asylum seekers in the past seven years sought asylum in the EU, with Germany, Austria, Hungary and Sweden being the top destinations. Afghans are the second largest group of first-time asylum applicants in Europe. For the period of the last 7 years, Germany has been the top country in which Afghans lodged asylum claims [4,5,6]. It seems that the number of Afghan refugees is also increasing in Switzerland. Indeed, a study carried out in the Emergency Department of the Inselspital showed an increase in the number of visits by Afghans in recent years [7,8,9].

It is therefore argued that it is of great importance to study and document the health status of refugee populations, in order to provide us with an overview of demographics, infectious diseases, metabolic and nutritional abnormalities, chronic cardiometabolic diseases, mental disorders and psychological distress, and to identify protective factors which could support key health concepts and good clinical practice. Given the high variety of specific diseases among migrant groups, the diversity of the origins of refugees and asylum seekers, and the prevalence of asylum seekers of Afghan nationality in the aforementioned countries, we chose to focus on this specific population. In the present integrative review, we describe the impact of the risk factors specific to locations and living conditions, how these may lead to illness, the most prevalent diseases, the health access of the female population, as well as their neuropsychological overload, and emphasise the need for further research on conflict-induced health status. The paramount aim is to optimise the medical approach and health services for everyone in the world, irrespective of their country of origin and financial background.

## 2. Methods

Electronic databases searched were Medline, PubMed, ScienceDirect, UNHCR, Eurostat, OCHA, International Organization of Migration (IOM), BMC Public Health, PLoS Med, OECD, Eurofound, PLoS One, and WHO. Reference lists of the relevant literature were also reviewed for further related studies. Keywords used in the search were terms that depicted the person or event of migration (refugee*, Afghan refugees*, health status*, internal conflicts*, migration*, asylum seekers*, policy and practice*, biopsychosocial stress*, epigenetics*, communicable diseases*, non-communicable disease*, war*, conflict*), the target population (Afghanistan*) and health-related outcomes (health experience, health issue, health problem, mental health, psychological, mental problems).

We included academic articles, literature reviews, as well as grey literature (think tanks, non-governmental organisation and government reports) written in English and published after 1994. This date was selected as it was consistent with a study on the first major wave of Afghan migration initiated by the first five years of Soviet occupation [1]. The review included studies of any design, involving adults and children of Afghan origin who had departed from their country for any reason, either voluntary or involuntary and who were settled in a host country. Articles were not excluded if they did not focus on Afghan immigrants. Articles that focused on people from different nations were included if they reported similarities in disease and psychological burden. Articles were included that focused on physical health issues and which considered the influence of immigration. Titles and abstracts of retrieved studies were reviewed to assess whether they met inclusion criteria. If inclusion was not immediately clear, full texts of articles were retrieved and reviewed.

## 3. Main Parameters Impacting Refugees’ Health

The health of refugees and asylum seekers is influenced by the poverty in their countries of origin [8,9,10,11], countries with low human development index and thus populations with impaired health status. Furthermore, access to health system services may be restricted prior to migration due to ongoing war, poverty and violent conflicts and this appears to be strongly associated with undiagnosed and poorly treated populations [12,13]. The main factors contributing to the observed low health status include acute and chronic malnutrition, low vaccination rates, low healthcare availability [7], and poor distribution of services for cancer prevention [14].

The conditions during migration may include transport in flimsy boats or closed containers [15]. This is why this population is vulnerable to contagious infectious diseases, accidental injuries, malnutrition and metabolic disturbances, not to mention hypothermia, acute exacerbations of respiratory, gastrointestinal and cardiovascular diseases, obstetric complications and gynaecological events, as well as mental and psychologic distress [16,17]. Moreover, during their travel or on arrival, they frequently spend long periods of time in detention centres and refugee camps with suboptimal hygiene conditions and poor access to health services [16,17].

Even though there are published data highlighting proper hygiene in most refugee centres, poor hygiene, precarious living conditions, sanitation, and living in cramped areas predispose the immigrants to several epidemic diseases, such as cholera, metabolic abnormalities, and toxic concentrations of lead [7].

In this context, the experience of stress is vital, where stress is defined as an event where homeostasis is threatened and then restored by the organism through behavioural and physiological mechanisms. Stressors can be physical or psychological and their importance depends on both their intensity and their duration. “Stress” comprises not only emotional stress from stressful life events, but also stress induced in the organism through extrinsic stressors (e.g., chemical warfare, malnutrition). The body undergoes many endocrinological and epigenetic changes, which prove to be detrimental for homeostasis, and impair immunoregulation in response to disease [10]. The experiences of war, fleeing for asylum, living in temporary mass refugee settlements, and resettling in other countries expose refugees to multiple threats/effects on their physical and psychological health and wellbeing.

In this respect, the epidemiological profile of the country of origin should be considered when approaching refugees or asylum seekers, in order to allocate patients to a specific screening program (ECDC). Indeed, studies investigating specific medical conditions demonstrate significant variability between different refugee populations from different countries [12]. For example, Europe receives refugees mainly from Syria, Afghanistan, Iraq, Eritrea, and Somalia [18]. Given the variety of specific diseases among migrant groups, the diversity of the origins of refugees and asylum seekers, and the increasing numbers of Afghan refugees, we shall focus on this specific population and describe their health status, in order to optimise our medical approach and management.

## 4. Infectious Diseases

A number of communicable diseases have been reported to spread between migrants and refugees living in the EU. These include respiratory tract infections, tuberculosis (TB), hepatitis A, typhoid fever, shigellosis, meningococcal meningitis, cutaneous diphtheria, scabies, and louse-borne relapsing fever [19].

A study examining the incidence of TB in the European Union/European Economic Area reported that of the 73,996 cases notified, 25% were of foreign origin [20]. Moreover, data from the National Surveillance System in the UK suggest that diagnosis of tuberculosis is delayed among persons of foreign origin [21]. Among the countries mentioned above, the reported incidence in Afghanistan is 189 new cases per 100,000 population [22]. Moreover, refugees living in detentions centres and camps, which are associated with overcrowding and poor ventilation, may be at increased risk of TB exposure [23]. In addition, refugee patients with TB may not complete treatment, leading to clinical relapses and drug resistance [24]. Furthermore, malnutrition is higher among refugees living in poor socioeconomic conditions and is associated with TB prevalence and its reactivation [25,26,27,28]. In a recent study, Kumar et al. (2020) found that the prevalence of latent TB infection among newly arriving refugees into the United States was significant high in Afghan population [29]. Hence, awareness for TB, strength of the surveillance strategies, compliance with measures to limit transmission and training of healthcare workers should be optimised. Screening would undoubtedly be impossible during exodus from the country, so screening at the point of entry would be more reasonable.

Infections of the gastrointestinal tract, and especially hepatitis A and typhoid fever, have previously been reported to be endemic in all of the above-mentioned countries [30], whereas multiple cholera outbreaks have been reported in Afghanistan [31]. A meta-analysis of the seroprevalence of chronic hepatitis B in immigrants from central and south Asia found intermediate seroprevalence [32]. Only a few studies have recorded the prevalence of hepatitis B in Afghanistan. Previous studies suggest that HBV prevalence is 6.15% in the overall population [33]. However, the prevalence among Afghan refugees may be significantly different [34]. Indeed, published data have shown that the prevalence of HBsAg among Afghan refugees varies from 4.1% in the United States to 60% in Dalaki, Iran [35]. Unfortunately, data are scarce on the overall prevalence of Hepatitis C virus and HIV in Afghanistan [36]. Given the higher incidence of HCC, liver cirrhosis, and the associated morbidity and mortality, screening strategies should be established.

Leishmaniasis affect 1–1.5 million people worldwide and is the ninth most prevalent infectious disease globally [37,38]. Afghanistan also continues to be one of the countries with the highest prevalence of leishmaniasis. In addition, it is also reported that Kabul, the capital of Afghanistan, is the largest focus of cutaneous leishmaniasis worldwide [39]. It is estimated that national incidence of leishmaniasis in Afghanistan is >200,000 cases, the majority of them being caused by Leishmania tropica and Leishmania major [40]. Various parameters such as political instability, poor healthcare infrastructure, dysfunction of the public health system, and ongoing conflict contribute to an increase in incidence. Moreover, epidemiological factors such as mass migration, overcrowding, incomplete treatment, and failure of vector control strategies may have led to the many outbreaks of the disease recorded in Afghanistan [37,39]. The data on the incidence of leishmaniasis among Afghan refugees are scarce. A systematic review and meta-analysis of the Afghan population in Iran reported a proportion of 7% for leishmaniasis [41].

Malaria is a common endemic disease in Afghanistan threatening about half of the Afghan population. Incidence has increased in recent years, especially in the Eastern regions [42]. In recent decades, the numbers of malaria cases has sharply increased in Afghanistan and—as with leishmaniasis—many outbreaks of the disease have been recorded [43]. The majority of the cases are caused by Plasmodium vivax (70–95%), followed by Plasmodium falciparum [44]. The data on the prevalence of malaria among Afghan refugees are limited, but an older study from Pakistan (2002) reported a tenfold increase in malaria cases between 1981 and 1991, that was probably associated with the arrival of 2.3 million Afghan refugees in Pakistan’s North West Frontier Province [45]. However, in subsequent years, the incidence of malaria in refugee camps decreased by 25% due to preventive measures, and this highlights the significance of control activities and prevention strategies [45]. In addition, in a systematic review and meta-analysis, Pourhossein et al. (2015) reported a high incidence of malaria amongst Afghan immigrants living in Iran and clearly showed the importance of monitoring the health status of immigrants in reducing the spread of communicable diseases [41].

The overall prevalence of intestinal parasitic infections in Afghanistan is high (39%), with higher reported rates in children and adolescents [46]. Commonly reported intestinal parasites in Afghan population include Ascaris lumbricoides, Giardia intestinalis, and Hymenolepsis nena [46]. Intestinal parasitic diseases are also common among Afghan refugee populations. Haq et al. studied the prevalence of Giardia intestinalis and Hymenoleps in a cross-sectional analysis of Special Immigrant Visa holders from Afghanistan into the United States and found high prevalence (30.7%) of at least one intestinal parasite. The most commonly detected parasites were Blastocystis, Giardia, and Dientamoeba [29]. Given that parasitic diseases may cause severe and lethal diseases [47], prompt prevention and eradication therapies should be established, especially for this vulnerable population.

Furthermore, antimicrobial resistance is a worldwide public health problem (World Health Organization. Global Action Plan). Research indicates that antimicrobial resistance is increasing in low-income countries with poor hygiene strategies and uncontrolled antibiotic use [48]. Moreover, there is increasing evidence that the mobile human population may contribute to the global spread of resistant bacteria [49]. A recent review of 238 articles related to antimicrobial resistance reported that the most frequent origin of travelers with drug resistant microorganisms was Asia, with 36% [49]. These findings are in accordance with the findings of Aro et al., who examined the prevalence of MDR-bacteria in refugees and asylum-seekers treated at Helsinki University Hospital. They reported colonisation in 45% of the cases, with remarkable differences between countries [50].

## 5. Non-Communicable Diseases

According to the WHO, non-communicable diseases (NCDs) are responsible for more than 70% of deaths worldwide [51]. Previous surveys have shown that NCDs cause between 33.3% and 37% of total deaths in Afghanistan [52,53].

The epidemiological profile of immigrants, refugees, and asylum-seekers is heterogeneous and differs from that of the population of the receiving country [54]. Most of the published studies have focused on communicable diseases, preventive care, pathological conditions associated with inequity in health, and healthcare access [54]. However, the prevalence of non-communicable diseases among refugee population is also high [55].

More specifically, some NCDs are more commonly associated with specific migrant populations and as previously discussed, frequently depend on the country of origin. It has been reported that the burden of NCDs is higher in adult refugees than in age-matched local born residents, probably because of the increased prevalence of chronic conditions in low-income countries [56]. Indeed, several studies have shown that refugees suffer from chronic conditions such as anaemia, arterial hypertension, malnutrition and its comorbidities, as well as impaired glycaemic control [7,57].

Exposure to the sun and vitamin D intake are environmental factors that have been widely associated with NCDs’ development and activity in the Afghan population as well. Vitamin D seems to be an intermediary between the two since UVB produces this vitamin under physiological mechanisms. The consumption of vitamin D enriched food appeared to have protective effects later in life for the risk of NCDs, thus suggesting an epidemiological link between the risk of NCDs and vitamin D [58,59]. Epidemiological and clinical studies have supported a potential role for vitamin D in maintaining balance in the immune system [60]. This is a vitamin deficiency can be promptly and easily corrected.

The prevalence of malignant diseases varies geographically and among nations. Non-western migrants are prone to malignancies related to infections experienced in early life, such as liver, cervical, and gastric cancer. In addition, those who originate from western nations frequently suffer from cancers related to a western lifestyle, such as colorectal, prostate, and breast cancer [61]. For Afghan refugees, the most common malignancies are gastrointestinal, haematological, genitourinary, breast, and in the CNS [62]. A retrospective cross-sectional study of 23,152 Afghan refugees reported that cancer was the second most common type of referral, with malignancies of the gastrointestinal track being the most frequently encountered cancer in Afghan patients between 18 and 59 years old with 13.3% [62].

Since it is well documented that malignancies are often underdiagnosed in low- and middle-income countries [7], a focus on the diagnosis and treatment of the specific diseases in refugees should be of high importance and priority.

Although environmental factors (i.e., lack of regular exercise, high-fat diet, and tobacco use related to war and lifestyle habits) might predispose to the development of malignancies and seem widespread among refugees [62], the data overall emphasise the need to develop prevention strategies, such as fecal occult blood testing, digestive tract endoscopy, mammography, and Pap tests, among refugee population, in order to prevent malignancies or to assure early diagnosis.

A previous study has shown that hypertension and cerebrovascular disease are observed more frequently in refugees and migrants from West Africa, while coronary heart disease is more common for refugees and migrants stemming from Northern Africa, Afghanistan, and Iraq [57,63,64]. Indeed, a study describing major health referrals in Afghan refugees in Iran reported 10% prevalence of heart disease [65]. The prevalence of tobacco consumption among Afghan men aged ≥ 15 years in Kabul has been estimated to be 35% [66]. The prevalence of risk factors of NCDs for persons older than 40 years has been found to be 13.3% for diabetes mellitus, 31.2% for obesity, and 46.2% for arterial hypertension [67]. In addition, cross-sectional studies have shown that the overall prevalence of overweight among the Afghan population is very high, ranging between 32.1% and 38% [68]. A recent study on the prevalence of risk factors of non-communicable diseases among Afghan refugees in Southern Iran showed that 94% of the study population consumed less than 5 servings of fruits/vegetables daily. Based on the same study, almost one-tenth of the participants consume tobacco, 5% of them are prediabetic, 20% have arterial hypertension, and almost 70% of them have some abnormality of the lipid status [69].

The high prevalence of heart disease among Afghan refugees could be additionally accentuated by the high incidence of post-traumatic stress disorder (PTSD) in this population [70]. In addition, this phenomenon is also observed in combat veterans returning from Afghanistan and Iraq. In fact, PTSD is a somatic disorder too and increases the risk of inflammation-related diseases and associated mortality. PTSD patients frequently suffer from obesity, arterial hypertension, hyperlipidaemia, diabetes mellitus, and cardiovascular disease [71]. Early diagnosis and the quality of the healthcare provided have been considered as possible additional factors for the higher mortality rates due to stroke in these populations [57,63,64]. Given that more than half of the Afghan population has at least one of the main risk factors for cardiovascular disease (i.e., obesity, hypertension, smoking, and elevated glucose levels) [67], prevention strategies, awareness raising, early detection, and sufficient management are essential to control non-communicable diseases among Afghan refugees.

## 6. Afghan Women and Access to Health Care

Afghanistan’s armed conflicts have undermined women’s access to healthcare for decades and continue to do so, aside from the sexual violence imposed on this gender. Afghan refugee women are a very vulnerable population, whose health needs have been neglected ever since the armed conflicts started taking place in 1979. A number of reasons have been cited as the obstacles toward adequate healthcare in women, such as financial struggles, health illiteracy, gender inequality, gender-based violence, and lack of autonomy. The limited access to healthcare and lower rates of antenatal care contribute to the higher incidence of adverse health outcomes, particularly during pregnancy, and could jeopardise the health of the mother and newborn [72].

Moderate to severe food insecurity has been documented, which could lead to adverse pregnancy outcomes in Afghan women suffering from lack of adequate nutrients and vitamins and may harm the health of the newborn. Several studies have shown that poor mental health has also been linked to the higher incidence of adverse pregnancy outcomes such as low birth weight or preterm birth. However, there is no evidence in this specific ethnic group [73].

The gender inequity is reflected in decision-making and power structures within families, based on culture and law that make it harder for women and girls to access healthcare [74]. Many women and girls even struggle to access basic information about their bodies, as well as important information they need to be able to make literate choices about their health, including their reproductive choices. Social stigma against discussing issues related to sexuality contributes to this, as do low rates of education and literacy among women and girls. However, the public health and education systems should also do better at providing this information [75].

Not only is sexual violence a substantial contributor to the ethnic cleansing and armed conflict that many refugees endure, but it is also a serious issue in refugee camps and in the human trafficking that refugees encounter on their journeys (mostly female but male and children too). It also significantly contributes to both physical and psychological impairment [53,76].

## 7. Neuropsychological Disorders

Epidemiological research indicates that, in addition to these common NCDs, refugees from low- and middle-income nations are at increased risk of mental, neurological, and substance abuse problems, including acute stress reactions, posttraumatic stress disorder, depression, epilepsy, and disorders associated with alcohol or substance use [77,78,79]. Indeed, the most common neurological disease among persons living in refugee camps and detention centers is epilepsy [80]. Moreover, the available literature highlights the fact that epilepsy is two to three times more prevalent in Sub-Saharan Africa than in high-income countries [81] and seems to be related to parasitic diseases (i.e., malaria, cysticercosis, onchocerciasis, toxocariasis, and toxoplasmosis), perinatal events, head injuries, HIV infection, malnutrition and other hereditary factors [82,83].

There are limited data on neuropsychiatric diseases among refugees from Central Asia [84,85,86]. The misconception of the public about neurological disorders and the difficulty of differentiating between neurological pathologies and psychiatric illness is an issue amongst the Afghan population. Social stigma is common for neurological and especially psychiatric patients. The incidence and prevalence of major neurological diseases in the Afghan population is unclear. The most often identified neurological conditions in Afghans are neuroinfections which could lead to epilepsy, developmental disabilities due to pregnancy-related complications such as hypoxic injuries, neuroimmunological diseases arising from inadequate nutrition, and spontaneous intracerebral haemorrhage [87].

The most common neuroinfections are viral encephalitis, tubercular meningitis and cerebral malaria. However, Afghanistan is one of only two countries in the world in which poliovirus type 1 is still circulating, regardless of the stupendous efforts to eradicate poliomyelitis (polio) from its population. Polio seriously threatens many Afghan children. This infectious viral disease is mostly spread through the fecal-oral route, prompted by inadequately sanitised surroundings. Paralysis is a threat once the disease moves to the nervous system. Since there is no cure for polio and vaccinations are not widely available for the Afghan population living in unhygienic conditions, prevention strategies are in shortage [88,89].

As mentioned above, malnutrition is a common phenomenon among refugees. Although malabsorption is generally considered to be a gastrointestinal problem, the effects of malabsorption extend far beyond the gastrointestinal tract and can include neurological dysfunction. Malabsorption may occur by a variety of mechanisms, both genetic and acquired, that interfere with the absorption of basic nutrients, vitamins, minerals, and trace elements.

Thiamine deficiency is associated with excessive use of or abrupt withdrawal from alcohol, poor or unbalanced diet, malabsorption, and increased excretion of the vitamin, causes typically seen in populations of developing and conflict-inflicted countries. Wernicke-Korsakoff syndrome (WKS) is a neuro-psychiatric manifestation of thiamine (vitamin B1) deficiency, also known as “wet brain” or alcoholic encephalopathy. It is generally known that Wernicke’s encephalopathy (WE) and Korsakoff’s psychosis are different stages of the same disorder. WE is an acute severe thiamine deficiency with a classic triad of confusion, ataxia, and ophthalmoplegia. Korsakoff’s psychosis is a chronic amnestic state that typically follows WE [90].

In reference to mental health issues, several studies have shown that refugees and migrants experience psychological distress [7,70,91,92,93,94,95], with depression and PTSD being the most common diagnoses [7,70] and which are additionally associated with infectious, neurological, and pulmonary diseases [7,96]. Increased rates of mental health problems in the specific population are attributed to several factors, such as trauma, culture shock, and language barriers. More specifically, Alemi et al. (2014) noted that trauma associated with the living conditions and experiences preceding flight are a possible cause for the vulnerability of Afghans to psychological distress [70].

In addition, the culture shock and the feeling of being uprooted that migrants and refugees experience when resettling in a new country seem to contribute to the long-standing nature of the observed mental health problems [70], with language playing a significant role, both in the maintenance of the symptoms and in the search for appropriate help [70,95]. It should be noted that males appear to be more prone to psychological disturbances, primarily because of socioeconomic factors (e.g., unemployment), which act as additional stressors and are perceived as identity threatening [70,95,97]. Altogether, this evidence highlights the challenges related to the mental healthcare of refugees and migrants and the need for improving the quality of services and the support systems offered. More studies on psychological distress in the specific population are required in order to provide healthcare professionals with the necessary information to approach such cases.

## 8. Policy and Practice Implications

Refugees have needs that differ from those of the host population, requiring effective care that recognises the impact of migration on physical and mental health. Health workforce interactions include personal engagement and relationships between health professionals and patients during the delivery of healthcare services. Refugee populations may also face challenges accessing healthcare that can shape their interactions with the host country’s health system and health workforce, including language and cultural barriers and lower levels of health literacy.

The migration experience, which—as already discussed—may involve poor transportation conditions, restrictive integration policies, and acculturation stress, can increase the vulnerability of refugees to both chronic and infectious diseases and influences the health status of individuals and their health needs. Key healthcare access barriers include language and cultural differences, low levels of health literacy and inadequate use of interpreting services.

Communication is of the utmost importance for effective provision of care to refugee and migrant populations and creates the foundation for trusting relationships between healthcare providers and patients. Continuity of care is also critical; however, the constant relocation of these populations may not allow service providers to have a full picture of their patient’s health history. This review adopts a specific approach, exploring the needs of refugee and migrants in relation to child and adult refugee populations’ health, reproductive health, mental health, preventive health, chronic diseases, as well as communicable and non-communicable diseases. A trauma-informed approach that is sensitive to previous experiences of human rights violations and other traumatic incidents is crucial for an effective approach towards refugees.

Further, unfamiliarity with the health system of a host country may lead to low usage of primary healthcare services by refugees and increase their presentations at emergency departments. Policy and legal frameworks in host countries may either create barriers for refugee populations accessing healthcare or support them in taking optimal health-related decisions.

Sexual health may be a sensitive topic for refugees to discuss with healthcare providers, which may be complicated further by low levels of health literacy and limited language proficiency. When discussing their sexual history with service providers, people from refugee and conflict-inflicted backgrounds may be reluctant to disclose experiences of sexual assault or gender-based violence. It is also important to consider gender preferences for interpreters, as well as whether or not to have partners or family members present for such discussions. Screening for sexually transmitted infections is absolutely essential for refugees and migrants, especially women who have fled high-risk environments for sexual violence.

Health literacy is highly relevant for refugees, who may have unique communication requirements that must be met in order to provide them with information to promote and maintain good health. In addition to general health literacy, education and training for healthcare providers in the host country are essential. However, it is important to note that other factors, such as different cultural beliefs, attitudes, and behaviours, have a significant impact on health. Nonetheless, low health literacy has been linked to a higher prevalence of health risk factors and comorbidities, lower participation in prevention activities, and increased hospital and emergency department admissions.

Traumatic, stressful, or difficult experiences in their home country, while migrating, seeking asylum, or settling in their host country, can lead to poorer mental health outcomes than the general population. Long-term, ongoing mental healthcare that extends beyond the period of resettlement has significant value in promoting the health of both the individual and their community, and mental health professionals’ ongoing support is encouraged. It is critical that healthcare professionals are aware of the specific mental health services and supports available to refugees and migrants, in order to make any necessary referrals and assist them in accessing treatment. The assessment, diagnosis, and treatment of mental health conditions can be aided by professional interpreters when necessary by adapting clinical guidelines for mental health conditions to particular contexts, training and supervising general health workers in their use, and engaging them when necessary.

Language-discordant consultations require the vital assistance of interpreting services. Patients are more likely to trust their healthcare practitioners if they have access to interpreting services.

In order to reduce the burden of disease and related risk factors, preventive health emphasises the need of immunisation, nutrition, and physical exercise. Due to under-immunisation, refugees are more likely to be vulnerable to childhood diseases that can be prevented by vaccination, particularly rubella, tetanus, and diphtheria. Through factors such as congestion, substandard housing, and insufficient hygiene services, the migration process can also expose people to much higher chances of developing infectious diseases such as HIV and tuberculosis, and more recently SARS-CoV-2 (COVID-19). In addition, even though some refugees may have had vaccinations in their home country, the vast majority do not have written records of their immunisation, so catch-up vaccination is advised in the absence of such records.

## 9. Conclusions

Four decades of civil war and serious human-rights abuses have forced millions of Afghans—men, women, and children—to flee their homes and seek refuge in other parts of Afghanistan or outside the country. Given the high variety of specific diseases among migrant groups, the diverse origin of refugees and asylum-seekers, and the increasing rates of Afghan refugees, the goal of the present literature review was to focus on the specific population and describe its health status in order to optimise medical approach and health services. A number of studies have addressed anaemia, hyperlipidaemia, arterial hypertension, diabetes, smoking, overweight, malnutrition, low socioeconomic status, and poor access to healthcare facilities as possible risk factors for non-communicable diseases among this vulnerable population. Gynaecological and obstetric pathologies and the reasons for their prevalence in this migrating group are not to be overlooked. Moreover, available data suggest that the Afghan refugee population is affected by infectious diseases, such as tuberculosis and other respiratory tract infections and parasitic diseases, as for example leishmaniasis, malaria and intestinal parasitic infections. Infectious, neurological, and pulmonary diseases are additionally associated with depression and PTSD, with the latter being enhanced by trauma, culture shock, and language barriers. However, data on most causes of mortality and morbidity among Afghan refugee population are scarce. Additional studies are needed in order to provide us with comprehensive, high-quality data about the health status of Afghan refugees in order to allow us to assist physicians and humanitarian agencies in serving and treating this vulnerable population.

## Data Availability

Not applicable.

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
