# Peer review of "Health Status of Afghan Refugees in Europe: Policy and Practice Implications for an Optimised Healthcare"

_ijerph, 2022, doi:10.3390/ijerph19159157_

Round 1
Reviewer 1 Report
Review ijerph-1770589 “Health Status of Afghan Refugees: A Review of the Literature”
Thank you for the opportunity to review this work, which examines common health issues among Afghan refugees with a view to assembling recommendations for medical treatment, and further research, for this population. Rather than being based on new empirical research, this paper draws together other existing studies to create a portfolio or primer specific to the Afghan refugee group.
The topic of refugee health is an important one for population/public health, as internationally this is a huge phenomenon where health issues/impacts span across multiple primary and secondary healthcare areas. Focus on the Afghan refugee sub-group is also important due to the long-standing nature of Afghan asylum-seeking and potential increases in refugee flows following the re-establishment of Taliban control over the country.
However, I do not feel that this piece as it stands has the definition, focus and breadth of thinking to be able to deliver the critical, comprehensive and clear overview for policy/practice implementation it proposes to provide. There are interesting findings summarised here, but very major revisions would need to be undertaken to refine the work to make it readable, useable and reliable. I recommend the authors take the work away, clarify and restructure the piece for stronger impact, and consider bringing in a social science co-author (e.g. anthropologist) to advise on strengthening some of the cultural awareness issues in the paper.
I make some specific comments for suggested revisions/reworkings of this paper below:
1. Introduction – I found this somewhat disjointed; the text introduced figures about Afghan refugee numbers and drivers worldwide/Europe-wide and then in two countries, then skipped to talking about the global refugee crisis generally, then to what seems to be a focus on stress and its physical and psychological manifestations, as a justification for why this review is necessary. I might suggest that this might work better in a different order – global refugee crisis, then details about the specific Afghan refugee group focused on here after that, narrowing the focus progressively. Suggest the detailed section about “stress” would be better positioned in Section 2; at this point it would be sufficient to state, with references, that the experiences of fleeing for asylum, living in temporary mass refugee settlements, and resettling in other countries expose refugees to multiple threats/effects on their physical and psychological health and wellbeing.
Be careful about contradictions; on line 71 the focus on Afghan refugees is justified by “increasing rates of Afghan refugees”, but earlier (line 44-45) it is stated that “figures have been on the decline in recent years”.
Be sure that the introduction sets out clearly what this paper will do, what sections/themes will follow and what its central argument is.
2. Main Parameters Impacting Refugees’ Health – This could be restructured into clearer sections as, again, it is somewhat disjointed. Consider three sections – conditions before fleeing homeland, conditions in transit/temporary settlements, conditions in new host country. Within each section have clear physical/mental wellbeing elements defined.
Regarding epidemiological profiling of refugees by country of origin (line 100) – I agree to an extent but what about when there are significant differences within a country’s refugee population in terms of lived experience, socioeconomic conditions and culture? Taking the population of refugees from Afghanistan as an example, this is made up of several distinct ethnic groups, including Pashtun who are the ethnic majority and are largely refugees from the war and destruction of their homes, and the Hazara ethnic minorities who are refugees from religious persecution and ethnic cleansing by the Taliban. These two groups, with two different lived experiences before, during and after fleeing Afghanistan, could not be argued to have similar epidemiological profiles.
3. Infectious Diseases and 4. Non-communicable Diseases – these sections also lack clarity, jumping from one disease type to another and between general and specific with no consistency. Three things should be established for each disease group – 1) Which ones are prevalent for this Afghanistan group (regardless of trends in comparison to other groups, that distracts from the argument); 2) Why are these diseases so prevalent for the Afghanistan group; and 3) What intervention or approach needs to be taken either at a health policy or practice level to screen, treat, protect against, etc, and at what stage of the refugee journey?
Acronyms e.g. for tuberculosis should be chosen and used consistently (either TB or TBC) and should be introduced in parentheses after the first use of the full term e.g. tuberculosis.
5. Afghan Women and Access to Healthcare – good to see a gender lens but be careful around what is presented here to take a culturally sensitive approach. For example, yes to the limited access of healthcare/health literacy impacting pregnancy/reproductive health, but being sensitive and patient-focused around culturally-informed reproductive health practices and attitudes and appropriate cultural liaison for interventions.
One other thing overlooked here are some of the secondary issues caused by experiences which may happen in the refugee journey – e.g. gender-based violence in the war or in the refugee camps – and the physiological and psychological health repercussions of this. In particular where you speak about “personal hygiene practices of women” (line 301) be careful to shape this as borne from the conditions of the refugee journey (e.g. bad access to sanitation/hygiene in camps) rather than what currently sounds quite condescending on a personal level.
6. Neuropsychological Disorders – needs a clearer distinction between Neurological and Psychiatric/Psychological; at the moment it swings between the two.
7. Conclusion – before this it would be good to have a clear section on Policy and Practice Recommendations – while some are suggested for each disease type/issue in the previous sections, consider drawing together a clear set of recommendations, first at a national/regional health policy level, then at a more healthcare practitioner/service level, for general things to establish, look out for, put in place etc.
Other – this piece would benefit from a thorough proof-read for expression and grammar by a third party, as the expression has quite a lot of quirks that decrease readability.
Author Response
Please see the attachment. Kind regards and thank you!

Reviewer 2 Report
Thanks for your article. Maybe you can add difference referances that have studied in other countries about refugees.
Reviewer 3 Report
The paper is an international literature review on refugee health with a particular focus on health within Afghanistan and among displaced Afghan peoples. Review of refugee health matters is useful, particularly pointing to gaps in research.
While the level of written English is good, there are enough minor errors and nuances that a thorough review by a native English speaker versed in scientific publications is recommended. Section 5 on women’s health stands apart as having clearer writing.
The methodology for this literature review is not described, so the appropriateness and comprehensiveness of the search criteria used cannot be assessed.
The very low literacy levels in this population have been mentioned (particularly in the section on Afghan women) but the impact of this on health literacy and health status could be emphasised elsewhere in the paper.
The references relating to refugee mental health are somewhat narrow. There is a huge body of evidence in this area, much of which would apply to the Afghan population as much as any other.
Specific comments (with line numbers noted):
L 77-81: acute health issues linked to fleeing (such as hypothermia) are mixed in with long term health issues for refugees
L91-98: repetitive from lines 66-69
L108 and elsewhere incl abstract: rather than saying “its health status, more colloquial to say “their health status” (even though referring to a population).
L111: implies malaria is contagious
L170: says “incidence of malaria of 40%”. Incidence should be expressed as a rate per population per time, not as a percentage.
L216 - 230: this detailed vitamin D metabolism information is unnecessary, and does not sit well with the rest of the paper. Paragraph should be deleted, except perhaps the last sentence.
L248: fecal occult blood testing is a far more appropriate method than endoscopy to detect bowel cancer
L265-7: beware overstating cause and effect between PTSD and cardiac disease etc. Association does not equal cause.
L318/19: there is reference to data on countries “in the Middle East” – important to note that Afghanistan is not part of the Middle East but rather Central Asia; so, these data may not be entirely relevant – just need to make the geographic distinction.
L322: stigmata is wrong word - stigma
L343: it is unclear why there is a paragraph about thiamine. It is just one of many nutrients that might be deficient and leading to neurological health issues eg zinc, vit B6, etc
Author Response
Please see the attachment. We really appreciated your valuable input! Kind regards.

Round 2
Reviewer 1 Report
Thank you to the authors for incorporating many of the suggestions of the previous round of reviews. Improvements have been made, and the structure is clearer - however, the contribution of this piece still remains unclear.
The authors have stated it is an integrative review, which is fine, and that their goal is to "identify protective factors which could ensure key health concepts and good clinical practice" (line 63-4). It should be made clearer where the information/sources are that are being drawn on; perhaps a brief "Methods" section? With regards to that goal of identifying protective factors and making clinical practice recommendations is what was meant by a "Policy and Practice Implications" section in the last review - this would not need new research, just summarising and drawing together key biopsychosocial factors and approaches that are common across the groups of diseases.
With regard to making a socioeconomic investigation of refugees - this was not the intent/recommendation. Merely making the point in the review, that could be stated also in the paper, that it is not possible to make generalisations across a diverse group like Afghan refugees, as not "everyone had to endure the same conditions while fleeing the country for example", either before or afterwards. Socioeconomic is one factor, but religion/ethnic minority status also has vast implications for different experiences, both in migration drivers and journeys. See https://dspace.mit.edu/handle/1721.1/97605 for some examples.
Where the authors say "We thought that gender-based violence would have been too critical/sensitive to mention. Are you instructing us to make a reference on sexual misconducts/violence?" - Yes absolutely, if this can be done with empathy and dignity. Sexual violence is not only a significant factor in the armed conflict and ethnic cleansing many refugees experience, but is also a major problem in refugee camps and in the human trafficking experiences of refugees journeys (mostly female but male and children also). It is also a major source of both physical and psychological trauma. If this paper seeks to inform health policy and guide healthcare practitioners it does a disservice to overlook this factor due to sensitivity concerns.
I think if this piece were tightened up - the reason for the review made more clear beyond just "this is a big problem for healthcare" and each themed section given a more structure separating issues and healthcare approaches before presenting overall healthcare policy and practice recommendations - it would make a really valuable contribution for healthcare practices, systems, training colleges and individual physicians to draw from. At the moment, however, it merely draws together some existing literature, without enough deeply critical thought, and profiles problems without clear solutions.
Reviewer 3 Report
The title of the article reads “/A Review of the Literature/”. My earlier comment “methodology for this literature review is not described” referred to the fact that literature reviews should include information on: search terms used, which databases searched, years searched, languages included, number of articles found, number excluded as not relevant, etc.
Overall, the English quality still needs review. For a written academic paper, there are aspects that can be improved. This includes the additional sentences added to this draft.
I have concerns about the sentence added in lines 298-299 which states "/The limited access to food and shelter during the migration *forces them to neglect* sanitation and hygiene. *Personal hygiene behaviors of women*./." The meaning of the first sentence is unclear. The start of the second sentence sounds derogatory towards women (although I'm sure is not intended to be) and needs changing. There is also no evidence that limited feminine hygiene leads to reproductive infections in women.
The focus on Wernicke’s encephalopathy, in particular, as a result of malnutrition is still a mystery. It stands out as an unusual paragraph.
